# Health Care for People with Disabilities in the Unified Health System in Brazil: A Scoping Review

**DOI:** 10.3390/ijerph19031472

**Published:** 2022-01-28

**Authors:** Márcia Andrea Oliveira da Cunha, Helena Fernandes Santos, Maria Eduarda Lima de Carvalho, Gabriella Morais Duarte Miranda, Maria do Socorro Veloso de Albuquerque, Raquel Santos de Oliveira, Adrião Filho Cavalcanti de Albuquerque, Loveday Penn-Kekana, Hannah Kuper, Tereza Maciel Lyra

**Affiliations:** 1Faculty of Medical Sciences, University of Pernambuco, Recife 50100-130, Brazil; marcia.oliveira@upe.br; 2Political Science Department, Federal University of Pernambuco, Recife 50070-460, Brazil; helenafsantos8@gmail.com; 3Post-Graduation in Public Health, Aggeu Magalhães Institute, FIOCRUZ, Recife 50670-420, Brazil; melc.duda@gmail.com; 4Academic Area of Public Health, Center for Medical Sciences, Federal University of Pernambuco, Recife 50070-460, Brazil; gabymduarte21@gmail.com (G.M.D.M.); or maria.valbuquerque@ufpe.br (M.d.S.V.d.A.); raquelsoliveira78@yahoo.com.br (R.S.d.O.); 5Department of Collective Health, Aggeu Magalhães Institute, FIOCRUZ, Recife 50670-420, Brazil; tereza.lyra@fiocruz.br; 6Recife Health Department, Recife 50080-000, Brazil; adriao@recife.pe.gov.br; 7Faculty of Epidemiology and Population Health, London School of Hygiene & Tropical Medicine, London WC1E 7HT, UK; loveday.penn-kekana@lshtm.ac.uk

**Keywords:** disability, healthcare access, health policy, Brazil

## Abstract

People with disabilities have greater need for healthcare on average, but often face barriers when accessing these services. The Brazilian government launched the National Health Policy for People with Disabilities (PNSPD) in 2002 to address this inequality. PNSPD has six areas of focus: quality of life, impairment prevention, comprehensive health care, organization and functioning of health services, information mechanisms, and training of human resources. The aim of this article was to undertake a scoping review to assess the evidence on the experience of people with disabilities in Brazil with respect to the six themes of the PNSPD. The scoping review included articles published between 2002 and 2019, from four electronic databases: PUBMED/MEDLINE, LILACS, Science Direct, and Scielo. In total, 8076 articles were identified, and after review of titles, abstracts, and full texts by two independent reviewers, 98 were deemed eligible for inclusion. The evidence was relatively limited in availability and scope. However, it consistently showed large gaps in delivery of healthcare to people with disabilities across the six dimensions considered. There was lack of actions aimed at promoting quality of life; insufficient professional training about disability; little evidence on the health profile of people with disabilities; large gaps in the availability of care due to widespread physical, informational, and attitudinal barriers; and poor distribution of the supply and integration of services. In conclusion, the policy framework in Brazil is supportive of the inclusion of people with disabilities in health services; however, large inequalities remain due to poor implementation of the policy into practice.

## 1. Introduction

People with disabilities include those who have long-term physical, mental, intellectual, or sensory impairments that, in interaction with various barriers, may hinder their full and effective participation in society on an equal basis with others (definition of the UN Convention of the Rights of Persons with Disabilities) [1]. Globally, there are at least 1 billion people with disabilities, making up 15% of the world’s population [2]. People with disabilities face widespread barriers and exclusions, including with respect to healthcare, and this contributes to their inequalities in health status and healthcare access [2,3]. It is therefore important to develop policies and programmes to support people with disabilities to overcome barriers, reduce inequalities, and realise their rights, including with respect to healthcare access.

Brazil can potentially provide a good practice example, as it has a progressive policy that should support the inclusion of people with disabilities in the health system. According to the Brazilian Federal Constitution of 1988, it is the duty of the State to take care of health and public assistance and the protection and guarantee of rights of people with disabilities [4]. Furthermore, in 2002 the National Health Policy for Persons with Disabilities (Política Nacional de Saúde da Pessoa com Deficiência-PNSPD) was implemented by the Ministry of Health after intensive lobbying from the disability community to support access to holistic healthcare for people with disabilities [5]. The PNSPCD had aims of universality, integrality, and equity, and it provides specific guidelines in the implementation of the health care policy relevant to people with disabilities. The PNSPD defined six areas of focus: (1) promotion of quality of life; (2) impairment prevention; (3) comprehensive health care; (4) organization and functioning of care services for persons with disabilities; (5) expansion and strengthening of information mechanisms; and (6) human resources training. This commitment to the right to healthcare for people with disabilities was reinforced by the ratification of the UNCRPD in Brazil in 2008 and again by the launching of the National Plan for the Rights of Persons with Disabilities—Living without Limits in 2011 [6]. The latter had the participation of more than 15 ministries and the National Council for the Rights of Persons with Disabilities (Conade), involved all federal entities, and had predicted investments of 7.6 billion reais (approximately 1.3 billion USD) until 2014 [6]. Specific commitments were also made to expand rehabilitation services and the provision of assistive technology [7]. This focus is important, as there are at least 17.3 million people with disabilities in Brazil, with a relatively even split of people with visual, hearing, lower-limb physical, upper-limb physical, or psychological impairments [8].

The policy framework in Brazil is therefore supportive of the right to healthcare for people with disabilities. However, good policy does not automatically translate into good practice and good outcomes. The effective implementation of a public policy depends on sufficient allocation of resources, intersectoral programmatic support, the inclusion of different interest groups in decision-making, and, perhaps most importantly, on the priority given by governments to the issue in question. These factors have been challenging in recent years in Brazil with the financial crisis and the shift towards neo-liberal politics. Yet, there has not been an assessment in Brazil of the realisation of the right to healthcare for people with disabilities, as set out in the PNSPD guidelines. We therefore undertook a scoping review to assess the evidence on the experience of people with disabilities in Brazil with respect to the six themes of the PNSPD guidelines: promotion of quality of life, impairment prevention, comprehensive health care, organization and functioning of health services, information mechanisms, and training of human resources.

## 2. Materials and Methods

A scoping review was undertaken to identify eligible original articles that provided evidence about healthcare for people with disabilities within the Brazilian health system. Eligible articles included qualitative or quantitative studies and were published in Portuguese, English, or Spanish between 2002 and 2019. Review articles, editorials, commentaries, and dissertations/theses were excluded. A protocol (PRISMA-ScR) was developed and published in the Open Science Framework (OSF. Available online: https://osf.io/8mghk/, accessed on 26 January 2022) [9].

Searches were performed in four electronic databases: PUBMED/MEDLINE, LILACS, Science Direct, and Scielo. The selection of descriptors was carried out through the Health Sciences Descriptors database—DeSC—and through MeSH (Medical Subject Headings). The following terms were used: Pessoas com deficiência, Pessoa com deficiência, Portador de deficiência, Portadores de deficiência, Pessoas com deficiências, Pessoa com deficiências, Portador de deficiências, Portadores de deficiências, Disabled Persons, People with Disabilities, People with Disability, Persons with Disabilities, Persons with Disability, Handicapped, Integralidade, Assistência Integral, Atenção à saúde, Atenção primária, Atenção secundária, Atenção terciária, and Atendimento Integral.

We undertook screening of titles, abstracts, and full texts to identify eligible articles. All screening was performed independently by two reviewers, with resolution of disagreements through consultation with a third reviewer at all stages. Before screening, 100 articles were selected to verify the degree of convergence between the reviewers, using the KAPPA index, which reached a value of 0.773, which indicated substantive agreement between the two reviewers [10].

Data were extracted from the eligible publications, including the methodological procedures of the studies, scope and location, type of study, research design, participant selection criteria, sample size and/or population, criteria for defining the sample size, period of data collection, characteristics of the studied population (gender, age, race, education, income), and levels of health care considered in the research. Narrative review was undertaken of the data to consider how health care for people with disabilities has been addressed in the scientific literature in Brazil, based on the guidelines of the PNSPD. Studies that addressed more than one category were grouped according to the objectives and main results presented.

Although in the PNSPD, the “Comprehensive health care for people with disabilities” and “Organization and operation of care services for people with disabilities” presented as separate axes [5], in this study, they were analysed together from the perspective of comprehensiveness of health care. This is because disability inclusion is understood to have two dimensions: (1) access to specific services offered in a well-defined health service facility and (2) access to comprehensive healthcare conceived in a broad way, which often takes place in a network of health services either because the various health technologies are distributed in a wide range of services or because the improvement of conditions of life is a task for an intersectoral effort.

## 3. Results

Overall, 8076 articles were identified (433 Science Direct, 684 Scielo, 2444 LILACS, and 4515 PUBMED/MEDLINE), of which 638 abstracts were screened, 221 full texts reviewed, and 98 articles selected for inclusion (5 PUBMED/MEDLINE articles, 27 Scielo articles, and 66 LILACS articles) (Figure 1).

### 3.1. Study Characteristics

The characteristics of included studies are described in Table 1. 

In terms of location, most studies were carried out at the municipal (40%) or local (39%) level. Almost all studies were undertaken in one of three regions: southeast (44%), northeast (33%), and South (18%). There was a relatively even split in studies focussed on primary (39%) or specialist (54%) healthcare. Studies focussed mostly on all disabilities (40%) or hearing (32%) impairments, while other categories received little attention. The study population was usually people with disabilities (55%) rather than healthcare professionals (26%). Most studies focussed on both males and females, and more studies addressed adults/older people rather than children. There was an even split between quantitative (48%) and qualitative (42%) approaches, and most studies used primary data sources (77%). The majority of studies provided participant selection criteria (81%). Sample size was generally rather small, as 61% of studies included 100 or fewer participants.

### 3.2. PNSPD Studies and Guidelines

“Comprehensive health care for people with disabilities” and “Organization and operation of care services for people with disabilities” sectors were combined and comprised 44% of all articles identified (Table 2). A quarter (25%) of articles focussed on promotion of quality of life for people with disabilities, while less literature was available on the topics of impairment prevention (12%), training (11%), and information mechanisms (8%).

### 3.3. Narrative Review of the Studies

#### 3.3.1. Promoting the Quality of Life of People with Disabilities

Twenty-four (25%) articles focussed on promotion of the quality of life of people with disabilities, including through provision of assistive technology (six articles); elimination of barriers that hinder the effective integration and inclusion (nine articles); inclusion and social participation (two articles); and evaluation of quality of life (five articles) [11,12,13,14,15,16,17,18,19,20,21,22,23,24,25,26,27,28,29,30,31,32,33,34]. Most of these studies took place at the primary care level and included a range of types of disabilities.

Six studies focussed on the importance of assistive technology to promote quality of life [11,12,21,28,31,32]. Of these, three studies highlighted the need to expand the availability of assistive technologies in the Public Health System (Sistema Unico de Saúde—SUS) and to offer equipment adapted to the specific demands of people with disabilities (wheelchairs, supply and replacement of hearing aids, devices for low vision, and equipment for the adaptation of children with cerebral palsy in everyday life) [11,12,21]. One study showed a positive correlation between the systematic use of hearing aids with the development of children’s hearing and language [11]. A final study highlighted the positive partnership between the State Government and the Public Ministry in offering wheelchairs in Natal, Rio Grande do Norte [12].

Nine articles addressed the barriers to the inclusion of people with disabilities in health services, and these studies were carried out at the primary care level, hospitals, and rehabilitation services. Lack of accessibility of health units was found to be an important barrier [11,13,14,17,18,19,20,23,34]. Two studies showed that hospitals and rehabilitation services were not physically accessible, and they highlighted the need for access ramps, door widening, signage, elevators, and handrails [12,20]. Studies at the primary care level paint an even grimmer picture. A nationwide study carried out in 41 municipalities with more than 100,000 inhabitants and containing 240 primary healthcare facilities concluded that 60% of these units are inadequate for the access of older people and people with disabilities [17]. Three studies focussed on the accessibility of health services to people with hearing impairment [47,102,106]. They showed that the lack of training and the non-adoption of the Brazilian Sign Language in primary care and specialized health units was the main barriers that limit their access to health services. Two further studies highlighted the barriers to accessibility of people with mobility impairments due to the lack of adapted public transport and the precarious conditions of the sidewalks [13,23].

People with disabilities are not a homogenous group, and three studies explored how intersectionality influenced the experiences in terms of quality of life [25,27,33]. A quantitative study [26] showed that quality of life of people with disabilities was negatively associated with difficulties with mobility, work, money, information, leisure, and sexual life. Poor people with disabilities may therefore be particularly vulnerable, and addressing healthcare barriers is only one component for improving quality of life. Another study confirmed these findings, indicating that the greater the social inequality, the lower the level of inclusion experienced by people with disabilities [25]. On the other hand, a third study highlighted the potential of health services to supporting quality of life [30]. It showed that children with cerebral palsy and consequently generally high care needs are more likely to be functionally independent and experience a good quality of life if they have good access to relevant services.

#### 3.3.2. Impairment Prevention

Impairment prevention was addressed in 12 studies [78,79,80,81,82,83,84,85,86,87,88,89]. These studies included consideration of neonatal hearing screening (NHS—nine studies); risk factors for deafness (one); early and late rehabilitation in deaf children and adolescents (one); early diagnosis of visual impairment in children (one); and prevention of visual impairment in people with diabetes (one). Research on the NHS was carried out in specialized outpatient services, rehabilitation services and hospitals/maternity hospitals [83]. The study showed that NHS coverage was below 25% in north-eastern Brazil, which is below the recommendations of the Ministry of Health. These levels should be raised, as there is positive correlation between the performance of NHS and the reduction of time to start early intervention and prevention of deafness. One study [86] explored the identification of risk factors for deafness and concluded that only 54% of physicians provide adequate guidance on the subject.

#### 3.3.3. Comprehensive Health Care and Organization and Operation of Care Services for People with Disabilities

Comprehensive health care and organization and operation of care services for people with disabilities was considered by 43 studies [35,36,37,38,39,40,41,42,43,44,45,46,47,48,49,50,51,52,53,54,55,56,57,58,59,60,61,62,63,64,65,66,67,68,69,70,71,72,73,74,75,76,77]. Nine studies examined access of people with disabilities to basic health services and identified a range of challenges, including low supply of basic services and specialized reference services for physical rehabilitation, insufficient organization of referrals and coordination of care, and lack of information about the care network for people with disabilities in addition to the lack of accessibility of facilities already described [38,39,41,46,47,64,65,71,77]. Consequently, people with disabilities had worse health outcomes, including a higher number of admissions to a psychiatric hospital and lower levels of breast self-exams and cervical cytopathological tests. One study considered specifically what changes are needed to primary care facilities to be able to meet the needs of people with disabilities and concluded that changes to structures and work processes are required [58]. Another study focussed on improvements needed in the coordination of health care and family and community guidance to improve care of children with disabilities in primary health settings [57]. Certain groups may face particular difficulties accessing health services, including older people with disabilities and people who are deaf, and they may need specific targeting with services and/or information [41,47,48]. A good practice example was also identified as provision of home visits for children with disabilities, which was shown to overcome some barriers and promote inclusion in health services [40].

A set of five papers focussed specifically on oral health care for people with disabilities at the primary health level, as the availability of these services is a specific commitment within SUS [38,45,46,64,66]. Again, people with disabilities experienced a range of difficulties in accessing these services due to low number of dental professionals with training and skills around disability, delay in scheduling appointments, and lack of general anaesthesia for specific cases. These health system failures resulted in worse oral health for people with disabilities; one study found that 76% of people with disabilities have difficulty in receiving care; 85% only seek urgent care; and 56% of dentists report difficulty in communicating with deaf patients [38].

The promotion of comprehensive health care within the context of specialized rehabilitation services and hospitals was the subject of 18 studies [35,37,43,48,49,50,51,53,54,56,60,61,62,66,67,68,72,73]. Of these, six focussed on the importance of greater integration of multidisciplinary teams in supporting people with disabilities [48,62,66,68,72,73]. Another survey highlighted the need for professionals encouraging recreational activities in order to improve the motor, cognitive, social, emotional, and language skills of children with physical disabilities [60]. An important concern with respect to the inclusion of people intellectual disabilities in rehabilitation services was the difficulty of accessing Psychosocial Care Centres (CAPS) and hospitalization [62]. Exclusions occurred because of lack of confidence of professionals around intellectual impairment that the absence of other reference services that could act as partners for such care.

Gaps in the availability of rehabilitation services were also noted. An evaluation of the degree of implementation of the rehabilitation network in Pernambuco involved eight municipalities with more than 100,000 inhabitants and 27 physical rehabilitation services [54]. It analysed 37 indicators in three dimensions (management, health care, and social control) and concluded that, although there is an expansion of services, the degree of implementation of this network in the state remains insufficient. Another survey limited to the city of Recife focussed on challenges to these services, which included low numbers of services/programs and multidisciplinary teams, lack of technological support, lack of coordination of care, and large gaps in availability of personnel, equipment, and information [50]. Three studies also revealed gaps in referrals and coordination of care between primary care services, rehabilitation services, and hospital units [49,65,75]. A central question seems to be whether rehabilitation services should be predominantly based in specialist centres or at the primary care level, as it is the first contact point for the community and supports the coordination of services and information.

Problems with quality of specialist care services for care of people with disabilities was shown through a study in Minas Gerais, with a quality index reaching only 25% in the items access, social needs, and information received [67]. Quality concerns may be more pronounced for some groups. For instance, two studies showed that people with spinal cord injury faced difficulties due to lack of professional qualifications to care for this group, fragmented care, difficulties in scheduling appointments and exams, difficulties in referral and counter-referral, and inconsistent care [25,65]. Another study highlighted similar problems in the preparation for discharge of people with disabling neurological injury, such as difficulty accessing rehabilitation services and lack of information about the Persons with Disabilities Care Network [49].

Qualitative research that considered the health care network for people with disabilities identified five major themes in the promotion of comprehensive care: rights, citizenship, education, transport, and leisure [24]. For these authors, the achievement of comprehensiveness depends essentially on intersectoral actions. For instance, the lack of accessible transport is a major barrier to accessing healthcare and needs to be improved. Another study confirmed this point, noting that while there were advances in the inclusion of people with disabilities in public policies in Brazil, the focus remained on charitable and biomedical approaches [77]. Social changes are therefore also needed to improve access to comprehensive healthcare for people with disabilities.

#### 3.3.4. Expansion and Strengthening of Information Mechanisms

Eight articles considered the availability and quality of information about disability [90,91,92,93,94,95,96,97]. These articles considered both the prevalence and type of disability as well as the living situation of people with disabilities. For instance, one survey collected information on the profile of people with disabilities assisted by the Family Health Strategy and showed that people with disabilities on average have low education and low income and that the most common impairments are due to neurological (43%), visual (12%) or hearing (9%) impairments, amputation (9%), or other causes (orthopaedic, mental, and malformations) [93]. The results also showed that most of this group use an auxiliary device and that only a minority (4%) receives physical therapy treatment. Little information is available from routine data sources. Only two articles collected information from SUS Health Information Systems. The first sought to understand the health needs of people with disabilities from medical records and information systems and showed that many were excluded from work and education and that they had few opportunities to access goods and services [91]. The second study aimed to describe the health profile of hearing-impaired people in the city of São Paulo through data collected at the SUS Department of Informatics [96]. Again, many challenges were identified, for instance, with living conditions, transport, negative attitudes, and experience of situations of violence. There is therefore a lack of information on the lives of people with disabilities, particularly from routine data sources, but available data consistently show large challenges and great unmet needs.

#### 3.3.5. Human Resource Training

Human resource training about disability was the focus of 11 articles [98,99,100,101,102,103,104,105,106,107,108], including promotion of skills to care for deaf people (six studies), training of Community Health Agents to register people with disabilities (one); training for assistance to blind people (one); and training to support people with any type of disability (three).

Three studies highlighted the need to include training on disability knowledge and skills in undergraduate health courses [100,101,102]. One article used qualitative methods to explore feelings of discomfort and unpreparedness of professionals in general to meet the needs of people with hearing loss [102]. It confirmed the lack of readiness of primary care professionals and supports the inclusion of disability in undergraduate courses in health disciplines.

Four articles considered communication barriers as the main difficulty in the delivery of care and the quality of the therapeutic relationship between professional and deaf user [100,101,107,108]. The importance of training of health professionals was highlighted, as they reported lacking confidence and preparedness to offer care to this group. They also highlighted good practice examples of individual initiatives of health professionals in training and improving care for people with disabilities, which are independent of public policies. In general, these articles emphasized adequate academic training and the effective implementation of public policies in promoting accessibility and social inclusion of people with disabilities.

## 4. Discussion

This scoping review of the evidence on the experience of people with disabilities in Brazil with respect to the six themes of the PNSPD guidelines highlights two major issues. First, there is a lack of data on the health and healthcare access of people with disabilities in Brazil. Few studies were identified, and those that were available were often small in scale, undertaken locally and in a few key regions (south, southeast, northeast), and focussed on only a few impairment groups. National and regional studies are lacking. Large gaps therefore exist, which need to be filled through further research with respect to groups (e.g., people with intellectual impairments), cross-cutting issues (e.g., gender, race, ethnicity, poverty, culture), geographic areas (e.g., central region, rural versus urban), and topic (e.g., the need for healthcare training and good practice examples). More evidence is needed on the inequalities in healthcare access facing people with disabilities so that appropriate interventions can be developed. In particular, a greater emphasis should be given to the use of routine sources of data for this purpose.

Second, although the policy framework in Brazil supports the inclusion of people with disabilities in healthcare, the available data show that this right does not seem to be realised, and large inequalities remain, as has also been noted by other researchers [17]. More efforts are needed to improve accessibility of primary healthcare facilities and linkages with specialist care. Notably, efforts are needed towards the training of healthcare workers on disability, which is specified under the PNSPD but not yet happening in reality. Health and healthcare access are important contributors to quality of life but not the sole determinants. Furthermore, healthcare access is not determined only by the health system, and factors such as poverty, attitudes, and accessible transport are also important. Attempts to improve healthcare access for people with disabilities therefore need to be intersectoral, linking and reinforcing strategies to improve living situations, transport systems, attitudes, and so on. Another important factor is that the inequities facing people with disabilities in Brazil are set against widespread inequalities in healthcare access in Brazil in general. More investigation is needed on the compounding role poverty plays in creating the health inequities for people with disabilities in Brazil and how they can be incorporated into other measures to level up healthcare access.

More efforts are therefore needed to implement evidence-based interventions to improve access to healthcare for people with disabilities in Brazil and thereby reduce inequalities. The PNSPD framework is helpful for guiding this action. These efforts for improvement can also be envisaged from a health system perspective [3]. The governance, meaning policies and laws, are already supportive of the inclusion of people with disabilities in the healthcare system. However, leadership, financing, and evidence need to be strengthened. Furthermore, changes are needed from the demand side (i.e., improving the autonomy, awareness, and affordability of healthcare by people with disabilities) and supply side (i.e., improving human resources availability and skills, accessibility of health facilities, and availability of specialist services). Together, these changes will improve health outputs for people with disabilities and consequently improve outcomes and reduce inequalities. However, more evidence needs to be gathered, both in Brazil and more globally, to document good practice and what works to improve disability-inclusive health.

There are a few limitations to this scoping review. A systematic review was not undertaken given the broad topic scope, and Scopus database was not included due to lack of subscription. Consequently, relevant papers may have been missed. Quality of papers was not considered although this is likely to have been low for many studies included. Furthermore, potentially relevant publications (e.g., theses, dissertations, grey literature) and publications not in English or Portuguese were excluded. There are issues with the generalisability of findings, as the studies included are mostly concentrated in the southeast and northeast regions and carried out at municipal and local levels although the objective was to analyse the national scenario regarding the health care of people with disabilities.

## 5. Conclusions

The policy framework in Brazil is supportive of the inclusion of people with disabilities in health services, and there has been some progress in the scaling up of rehabilitation services. However, large inequalities remain due to poor implementation of the policy into practice. In particular, there are issues with accessibility of services, lack of coordination of care, poor healthcare worker knowledge, and lack of information on disability, which collectively weaken the realisation of the right to health and perpetuate health inequalities. There is an urgent need for the Brazilian State to implement and expand the strategic actions of the PNSPD, promoting the quality of life of this population to repair the history of inequality of opportunities for these people.

## Figures and Tables

**Figure 1 ijerph-19-01472-f001:**
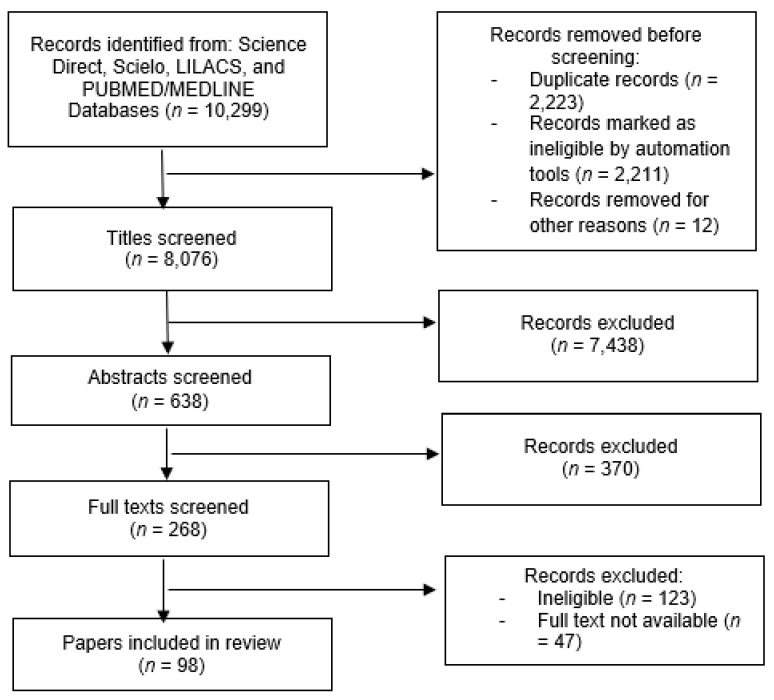
Study selection PRISMA flow diagram.

**Table 1 ijerph-19-01472-t001:** Characteristics of studies included in the scoping review.

Variable	Level	Percentage of Eligible Studies (*n* = 98)
Study location	National	2%
	Regional/district	4%
	State	15%
	Local	39%
	Municipal	40%
Region	Southeast	44%
	Northeast	33%
	South	18%
	Mixed/other	5%
Healthcare level	Primary healthcare	39%
	Specialist services	54%
	Mixed/other	7%
Disability type	All	40%
	Hearing	32%
	Visual	5%
	Intellectual	3%
	Multiple	2%
Participant type	Person with disability	55%
	Health professional	26%
	Health unit	7%
	Mixed/other	12%
Participant age	Child/adolescent	16%
	Adult/elderly	29%
	Mixed	12%
	Not applicable	43%
Participant gender	Mixed	63%
	Female only	3%
	Not applicable	34%
Study method	Quantitative	48%
	Qualitative	42%
	Mixed	10%
Data collection	Primary	77%
	Secondary	16%
	Mixed	7%
Sample size	1–100	61%
	101–1000	26%
	1001+	10%
	NA	3%
Participant selection criteria specified	Yes	81%
	No/NA	19%

**Table 2 ijerph-19-01472-t002:** Studies grouped according to PNSPD guidelines.

PNSPD Guidelines	Studies	References
Promotion of quality of life for people with disabilities	25%	[11,12,13,14,15,16,17,18,19,20,21,22,23,24,25,26,27,28,29,30,31,32,33,34]
Comprehensive health care for people with disabilities and organization and operation of care services for people with disabilities	44%	[35,36,37,38,39,40,41,42,43,44,45,46,47,48,49,50,51,52,53,54,55,56,57,58,59,60,61,62,63,64,65,66,67,68,69,70,71,72,73,74,75,76,77]
Impairment prevention	12%	[78,79,80,81,82,83,84,85,86,87,88,89]
Expansion and strengthening of information mechanisms	8%	[90,91,92,93,94,95,96,97]
Training human resources for assistance to people with disabilities	11%	[98,99,100,101,102,103,104,105,106,107,108]

## Data Availability

Not applicable.

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
