# Peer review of "Health Care for People with Disabilities in the Unified Health System in Brazil: A Scoping Review"

_ijerph, 2022, doi:10.3390/ijerph19031472_

Round 1

Reviewer 1 Report

This article presents a scoping review of health care for people with disabilities in the Unified Health System in Brazil.  Authors assert a need for this study due to the fact that people with disabilities often face barriers to healthcare services despite having a greater need for healthcare.  Authors review articles from a national database to explore the experience of people with disabilities in Brazil with regard to the six National Health Policy for People with Disabilities (PNSPD) focus areas: “quality of life, impairment prevention, comprehensive health care, organization and functioning of health services, information mechanisms and training of human resource”.  Feedback is as follows:

  1. Line 47- When authors note “ there are approximately 1 billion people with disabilities”, there should be a background and definition as to what constitutes a disability. Furthermore, there should be specification of the different types of disabilities (e.g., physical, cognitive/intellectual,  psychiatric, visual, hearing, etc.).  It would be good to see a breakdown of the numbers both globally and within Brazil.
  2. In the Introduction, authors establish a good rationale for this study along with how this study adds to the literature when explaining that there has yet to be an assessment of the right to healthcare for people with disabilities, as it relates to PNSPD guidelines.
  3. Line 186-190. Good mention that people with disabilities are not a homogenous group. Are there disparities in the experience of people with disabilities in Brazil according to factors such as race, ethnicity, geographic location, and gender?  These questions may be directions for future research.
  4. Lines 232-240 – Authors make an important mention of the need to assess the experiences of people with disabilities as it relates to oral health care needs.
  5. Line 306 – Good mention of human resource training and consideration of communication barriers. Was there any insight into effect of cultural factors/intercultural influences on communication barriers?
  6. Future considerations may also include effect of diversity, culture, and disparities as it relates to people with disabilities in Brazil.

Overall, this is an insightful, pertinent, and unique study.  Attending to some clarifying questions may help to improve the overall paper.

Author Response

Reviewer 1

  1. Line 47- When authors note “ there are approximately 1 billion people with disabilities”, there should be a background and definition as to what constitutes a disability. Furthermore, there should be specification of the different types of disabilities (e.g., physical, cognitive/intellectual,  psychiatric, visual, hearing, etc.).  It would be good to see a breakdown of the numbers both globally and within Brazil.

Response: We have included a definition of disability in the opening paragraph of the introduction. The breakdown of these numbers are not available globally. We have added (line 80): “This focus is important as there are at least 17.3 million people with disabilities in Brazil, with a relatively even split of people with visual, hearing, lower limb physical, upper limb physical or psychological impairments. [8]” We would prefer not to include more precise estimates given concerns with the source data.

  1. In the Introduction, authors establish a good rationale for this study along with how this study adds to the literature when explaining that there has yet to be an assessment of the right to healthcare for people with disabilities, as it relates to PNSPD guidelines.

Response: Thank you.

  1. Line 186-190. Good mention that people with disabilities are not a homogenous group. Are there disparities in the experience of people with disabilities in Brazil according to factors such as race, ethnicity, geographic location, and gender?  These questions may be directions for future research.

Response: There is limited data (cited lines 228 onwards) that people with disabilities who are poorer face more barriers. We did not identify data by ethnicity or geographic location and this is now cited as a limitation (line 426). 

  1. Lines 232-240 – Authors make an important mention of the need to assess the experiences of people with disabilities as it relates to oral health care needs.

Response: Thank you.

  1. Future considerations may also include effect of diversity, culture, and disparities as it relates to people with disabilities in Brazil.

Response: These points are now highlighted in line 426

  1. Line 306 – Good mention of human resource training and consideration of communication barriers. Was there any insight into effect of cultural factors/intercultural influences on communication barriers?

Response: We reviewed the papers again and could not find insights into effect of cultural factors/intercultural influences on communication barriers.

Reviewer 2 Report

This is a well written paper. The authors clearly described the background and procedures used to conduct the scoping review. The results were well organized and informative. Based on reviewed articles, the authors successfully identified the limitations of existing literature and the need to implement evidence-based interventions to improve access to care among people with disabilities in Brazil. 

The authors used four electronic databases for literature searches. However, it is unclear why the Scopus database was not included.

It appears that few articles listed in the references section were not mentioned in the narrative (#110 - #112, #114 - #115).

Author Response

  1. The authors used four electronic databases for literature searches. However, it is unclear why the Scopus database was not included.

Response: The Brazilian authors leading the review did not have a subscription to Scopus. Lack of inclusion of Scopus is now cited as a limitation (line 471).

  1. It appears that few articles listed in the references section were not mentioned in the narrative (#110 - #112, #114 - #115).

Response: These references (from 109 onwards) were incorrectly included from an earlier draft and have now been omitted.

Reviewer 3 Report

This is an important and useful addition to material concerning the inequities in additional barriers to accessing care for people with disabilities, in this case in Brazil. While generally well written, the paper would benefit from a more focussed introduction on the background to the National Health Policy  for People with Disabilities (PNSPD), for example what led to its introduction. 

While the material selected and search terms seem appropriate and the authors comment on the wider determinants (indeed, are there other policies which were introduced at the time of the PNSPD in say the education or housing sectors which may allude to a wider agenda?), there is no discussion of the inequities in access that exist for the population as a whole. I think some reference to this which I would expect to highlight the additional extent that certain disabilities may create.

Finally, the authors make a very useful observation about training of healthcare workers to address this. Is there any developments that may have occurred in this area which alludes to good practice and why?

Author Response

  1. While generally well written, the paper would benefit from a more focussed introduction on the background to the National Health Policy  for People with Disabilities (PNSPD), for example what led to its introduction. 

Response: More specificity on this point has been added to the introduction

“Brazil can potentially provide a good practice example as as it has a progressive policy which should support the inclusion of people with disabilities in the health system. According to the Brazilian Federal Constitution of 1988, it is the duty of the State to take care of health and public assistance, the protection and guarantee of people with disabilities [4]. Furthermore, in 2002 the National Health Policy for Persons with Disabilities (Política Nacional de Saúde da Pessoa com Deficiência - PNSPD) was implemented by the Ministry of Health after intensive lobbying from the disability community to support access to holistic healthcare for people with disabilities [5]. The PNSPCD had aims of universality, integrality and equity and it provides specific guidelines in the implementation of the health care policy relevant to people with disabilities”

Plus “These factors have been challenging in recent years in Brazil with the financial crisis and the shift towards neo-liberal politics.” (line 88)

  1. While the material selected and search terms seem appropriate and the authors comment on the wider determinants (indeed, are there other policies which were introduced at the time of the PNSPD in say the education or housing sectors which may allude to a wider agenda?), there is no discussion of the inequities in access that exist for the population as a whole. I think some reference to this which I would expect to highlight the additional extent that certain disabilities may create.

Response: We have now specified in line 443: “Another important factor is that the inequities facing people with disabilities in Brazil are set against widespread inequalities in healthcare access in Brazil in general. More investigation is needed on the compounding role poverty plays in creating the health inequities for people with disabilities in Brazil and how they can be incorporated into other measures to level up healthcare access.”

  1. Finally, the authors make a very useful observation about training of healthcare workers to address this. Is there any developments that may have occurred in this area which alludes to good practice and why?

Response: Although the training is within the guidelines of the PNSPCD, in practice, there is no action to train health professionals. We have highlighted the need for good practice examples in the limitations (line 360) and “Notably, efforts are needed towards the training of healthcare workers on disability, which is specified under the PNSPD but not yet happening in reality.” (line 436)